# Patterns of beverage purchases amongst British households: A latent class analysis

Nicolas Berger[1,2]*, Steven Cummins[1], Alexander Allen[3], Richard D. Smith[3,4], Laura Cornelsen[1]

1 Population Health Innovation Lab, Department of Public Health, Environments and Society, London School of Hygiene & Tropical Medicine, London, United Kingdom, 2 Sciensano, Brussels, Belgium, 3 Faculty of Public Health and Policy, London School of Hygiene & Tropical Medicine, London, United Kingdom, 4 College of Medicine and Health, University of Exeter, Exeter, United Kingdom

* Nicolas.Berger@lshtm.ac.uk

**Data Availability Statement:** The terms of our data access agreement mean that we cannot share data. However, data are available directly from Kantar Worldpanel (see www.kantarworldpanel.com/en for contact details). Our contact was Alex

## Abstract

### Background

Beverages, especially sugar-sweetened beverages (SSBs), have been increasingly subject to policies aimed at reducing their consumption as part of measures to tackle obesity. However, precision targeting of policies is difficult as information on what types of consumers they might affect, and to what degree, is missing. We fill this gap by creating a typology of beverage consumers in Great Britain (GB) based on observed beverage purchasing behaviour to determine what distinct types of beverage consumers exist, and what their socio-demographic (household) characteristics, dietary behaviours, and weight status are.

### Methods and findings

We used cross-sectional latent class analysis to characterise patterns of beverage purchases. We used data from the 2016 GB Kantar Fast-Moving Consumer Goods (FMCG) panel, a large representative household purchase panel of food and beverages brought home, and restricted our analyses to consumers who purchase beverages regularly (i.e., >52 l per household member annually) (n = 8,675). Six categories of beverages were used to classify households into latent classes: SSBs; diet beverages; fruit juices and milk-based beverages; beer and cider; wine; and bottled water. Multinomial logistic regression and linear regression were used to relate class membership to household characteristics, self-reported weight status, and other dietary behaviours, derived from GB Kantar FMCG. Seven latent classes were identified, characterised primarily by higher purchases of 1 or 2 categories of beverages: 'SSB' (18% of the sample; median SSB volume = 49.4 l/household member/year; median diet beverage volume = 38.0 l), 'Diet' (16%; median diet beverage volume = 94.4 l), 'Fruit & Milk' (6%; median fruit juice/milk-based beverage volume = 30.0 l), 'Beer & Cider' (7%; median beer and cider volume = 36.3 l; median diet beverage volume = 55.6 l), 'Wine' (18%; median wine volume = 25.5 l; median diet beverage volume = 34.3 l), 'Water' (4%; median water volume = 46.9 l), and 'Diverse' (30%; diversity of purchases, including median SSB volume = 22.4 l). Income was positively associated with being classified in the Diverse class, whereas low social grade was more likely for households in the

Rowberry: Alex.Rowberry@KantarWorldpanel.com.

**Funding:** NB and LC are supported by the UK Medical Research Council (grant number MR/P021999/1). The funder had no role in study design, collection, analysis, and interpretation of data, writing the report, and the decision to submit the report for publication.

**Competing interests:** The authors have declared that no competing interests exist.

**Abbreviations:** BIC, Bayesian information criterion; BLRT, bootstrap likelihood ratio test; BMI, body mass index; FMCG, Fast-Moving Consumer Goods; GB, Great Britain; LCA, latent class analysis; NSP, non-starch polysaccharide; RRR, relative risk ratio; SDIL, Soft Drinks Industry Levy; SES, socio-economic status; SSB, sugar-sweetened beverage; VLMR-LRT, Vuong–Lo–Mendell–Rubin adjusted likelihood ratio test.

classes SSB, Diet, and Beer & Cider. Obesity (BMI > 30 kg/m$^2$) was more prevalent in the class Diet (41.2%, 95% CI 37.7%–44.7%) despite households obtaining little energy from beverages in that class (17.9 kcal/household member/day, 95% CI 16.2–19.7). Overweight/obesity (BMI > 25 kg/m$^2$) was above average in the class SSB (66.8%, 95% CI 63.7%–69.9%). When looking at all groceries, households from the class SSB had higher total energy purchases (1,943.6 kcal/household member/day, 95% CI 1,901.7–1,985.6), a smaller proportion of energy from fruits and vegetables (6.0%, 95% CI 5.8%–6.3%), and a greater proportion of energy from less healthy food and beverages (54.6%, 95% CI 54.0%–55.1%) than other classes. A greater proportion of energy from sweet snacks was observed for households in the classes SSB (18.5%, 95% CI 18.1%–19.0%) and Diet (18.8%, 95% CI 18.3%–19.3%). The main limitation of our analyses, in common with other studies, is that our data do not include information on food and beverage purchases that are consumed outside the home.

## Conclusions

Amongst households that regularly purchase beverages, those that mainly purchased high volumes of SSBs or diet beverages were at greater risk of obesity and tended to purchase less healthy foods, including a high proportion of energy from sweet snacks. These households might additionally benefit from policies targeting unhealthy foods, such as sweet snacks, as a way of reducing excess energy intake.

### Author summary

#### Why was this study done?

- Policies to tackle obesity have increasingly targeted drinks, in particular sugary drinks, as a major source of excess sugar and energy.

- An important limitation of the current evidence is that policy targets are based on the 'average' household or individual, and do not take into account differences in households based on buying behaviours.

- Accounting for differences between households is important in understanding whether some people with obesity are effectively targeted by current policies, and whether policies are likely to have the intended effect on regular purchasers.

#### What did the researchers do and find?

- We applied a data-driven method, known as latent class analysis, to food and beverage purchase data from 8,675 British households, to identify population subgroups with similar patterns of drink consumption in order to identify high-risk households that could be targets for interventions.

- We found that 48% of households purchased medium-to-high volumes of sugary drinks and 16% of households purchased high volumes of diet drinks. Other households mainly purchased fruit juice, water, or alcoholic drinks such as beer or wine.

- Households purchasing high volumes of sugary or diet drinks were more likely to have low socio-economic status, higher BMI, and overall less healthy food purchases, characterised by a high proportion of energy obtained from sweet snacks (approximately 18%).

### What do these findings mean?

- Our results suggest that households at risk of obesity that purchase high volumes of sugary or diet drinks also have high purchases of sweet snacks.

- These households might additionally benefit from policies targeting unhealthy foods, such as sweet snacks, as a way of reducing excess energy intake.

## Introduction

Beverages, such as sugar-sweetened beverages (SSBs), fruit juices, and alcohol, are an important source of excess sugar and energy intake globally [1–4]. Recent evidence indicates consistent associations between SSB consumption and body mass index (BMI) [5], diabetes [3], and dental caries [6], and between fruit juice consumption and dental caries [4]. In Great Britain (GB), the National Diet and Nutrition Survey (NDNS) shows that, on average, SSBs contribute 11% of free sugar intake in adults and 20% in children aged 11–18 years [7]. Several studies in GB and elsewhere have found associations between socio-economic status (SES) and beverage consumption, with lower SES groups consuming more SSBs than higher SES groups [8–10].

The World Health Organization has recently recommended the use of fiscal policies to address poor diet, obesity, and related non-communicable diseases [11]. These policies include tools such as taxes and subsidies to improve economic access to healthy dietary choices, and create financial incentives for behaviours associated with improved health outcomes and financial disincentives to discourage the consumption of less healthy options. Recently, a growing number of countries have implemented fiscal policies aimed at reducing intake of SSBs and other beverages such as juices, energy drinks, and alcohol [12]. Some countries have supplemented these with product reformulation strategies [13]. In UK, for example, the Soft Drinks Industry Levy (SDIL) came into effect in April 2018, with an 18-pence and 24-pence tax per litre on beverages containing 5 or more and 8 or more grams of sugar per 100 millilitres, respectively. In addition to lowering demand for SSBs, the 2-tiered levy has encouraged the industry to reduce the sugar content of beverages, resulting in a decrease in the volume of sugars sold in beverages by 30% between 2015 and 2018 [14,15].

Policy interventions are, however, often heterogeneous in their effects [16]. Part of the success of fiscal and reformulation policies directed at beverages relies on their ability to affect high-risk individuals. That is, they should ideally reach individuals who drink targeted beverages in greater quantity and those at greater risk of overweight, obesity, and diet-related chronic diseases, and ideally also contribute to reducing socio-economic inequalities. An

important limitation of the current evidence is the absence of detailed information on individual consumer patterns of overall food and beverage purchases beyond population averages [17], in particular for high-volume consumers of beverages. This type of information is crucial to understand whether some high-risk individuals are effectively excluded from current or proposed policies, and whether policies are likely to have the intended effect on regular purchasers. For example, relatively little is known about what other kinds of food and beverages high SSB consumers purchase; this is important as it could determine whether they would be more or less likely to replace SSB purchases with other high-sugar food or beverages. While some of this information might be obtained from modelling studies or price elasticity analyses, these are usually restricted to pre-defined population subgroups, such as those based on income or weight status [18]. Categorising high-volume consumers on the basis of actual beverage purchasing behaviour may therefore help us better understand whether distinct types of beverage consumers exist, what their socio-demographic (household) and health characteristics are, and what kinds of policies might have the most impact on the energy intake of identified high-risk individuals.

This paper aims to provide a typology of regular beverage consumers on the basis of household purchasing behaviours. To do so, we use latent class analysis (LCA), a data-driven method that can help identify population subgroups with similar patterns of consumption and therefore identify high-risk households that could be the target for interventions [19–21]. We then explore whether identified types of beverage consumers can be characterised by household characteristics and BMI status, and whether beverage consumer types are indicative of the nutritional quality of all food and beverages purchased. To do so, we use large-scale product-level data from the 2016 GB Kantar Fast-Moving Consumer Goods (FMCG) panel, a household panel of food and beverages bought by British households and brought into their home. Although this does not cover purchases that are consumed outside the home (e.g., in restaurants), this type of data is preferred over dietary surveys because it contains highly disaggregated information on beverages and it is less vulnerable to underreporting and seasonality biases [22].

## Methods

This study is reported as per the Strengthening the Reporting of Observational Studies in Epidemiology (STROBE) guideline (S1 Appendix).

### Study sample

The GB Kantar FMCG panel is a representative consumer panel of food and beverages purchased by households in GB (i.e., England, Wales, and Scotland) and brought into their home. We obtained transaction-level data reported by 32,110 households between 4 January 2016 and 1 January 2017 (52 weeks) and self-reported socio-demographic data collected by GB Kantar FMCG. Participants scanned take-home purchases using hand-held barcode scanners. GB Kantar FMCG provided nutritional data on products through direct measurement in outlets, or using product images supplied by Brandbank, a third-party supplier. Where GB Kantar FMCG was unable to gather direct information, nutritional values were copied across from products with identical composition (6% of purchases for information on energy, for example), or an average value for the category or product type was calculated and used instead (12% of purchases for information on energy). The dataset has been described in detail elsewhere [23].

We restricted our analyses to households that consistently participated in the panel and reported purchases during the year (i.e., >14 days of purchases in every quarter) to exclude

data from households with intermittent participation in the panel [24], and to avoid seasonality bias. We further restricted our analyses to households that regularly purchased beverages (>52 l per year per household member) because they are more likely to be the target of and to benefit from policies addressing unhealthy beverage intake. Households purchasing extreme quantities of each beverage type were excluded (>364 l of each beverage type or >500 l in total per year per household member) to avoid potential bias caused by mass purchases made by some panellists, which would fall outside usual household consumption. The final sample included 8,675 households reporting 14,007,226 product-level purchases of food and beverages. Households that regularly purchased beverages represented 69% of panellists who consistently reported data (Fig 1).

## Measures

We calculated volume purchased from 6 beverage categories: SSBs (sugary beverages; juices and squashes with added sugar); diet beverages (beverages, juices, squashes, and flavoured waters branded as 'diet', 'low calorie', or 'no added sugar' including no more than traces of added sugar); fruit juices and milk-based beverages (including 100% fruit juices and flavoured milks such as chocolate milk); beer and cider; wine; and bottled water (including sparking water) (see details in S2 Appendix) [25]. Spirits, breakfast beverages, hot chocolate, tea, and coffee were not included because they were considered as less likely to be substitutes for the selected beverages. Results from analyses based on the 6 continuous variables did not converge towards an optimal number of latent classes (i.e., adding more latent classes continuously improved model fit until the models became too complex to converge); following previous research we therefore used volume tertiles in the LCA [26].

For each household, we also computed overall mean daily energy and nutrients purchased per household member from all food and beverages, without adjustment for age or sex [27]. We then calculated energy obtained from the 6 beverage categories combined, and from non-alcoholic beverages separately (SSBs, diet beverages, fruit juices and milk-based beverages, and water). We additionally calculated the energy obtained from selected food groups of public health interest. These are 'less healthy' food and beverages, identified using the UK Department of Health and Social Care nutrient profiling model [23,28,29], sweet snacks (a subset of the 'less healthy' category, including chocolates, confectionery, puddings, and biscuits), and fruits and vegetables (see S2 Appendix) [23]. To explore household differences in total energy purchased by food group, we used percent of total energy obtained from each food group and total fat, saturated fat, protein, carbohydrates, sugar, non-starch polysaccharide (NSP) fibre, and sodium from food and beverages purchased, measured as grams/1,000 kcal purchased. We used these relative nutritional variables (except for total energy) because absolute values were underestimated by about 3% on average (which could differ by latent class and socio-demographic characteristics) [23], due to households not reporting all purchases made throughout the year.

Available household socio-demographic variables provided by GB Kantar FMCG were as follows: region, life stage, occupational social grade of the main shopper [30,31], income, and self-reported BMI of the main shopper. Life stage was categorised by GB Kantar FMCG as pre-family (main shopper under 45 years, no children), young family (youngest child 0–4 years), middle family (youngest child 5–9 years), older family (youngest child 10+ years), older dependents (main shopper 45+ years, no children, 3+ adults), empty nesters (main shopper 45–64 years, no children), and retired (main shopper 65+ years, no children). Occupational social grade was categorised as follows: higher and intermediate managerial, administrative, or professional occupations (A&B); supervisory, clerical, and junior managerial administrative or

> **Available sample:** 32,110 active households reporting 37,963,159 product-level purchases of food and beverages between 4th January 2016 and 1st January 2017 (52 weeks)

> **Exclusion** of 1,891,753 food products with inconsistent nutritional information despite correction (5%):
>     943,322 desserts (e.g. mince pies, small tarts, cakes)
>     483,521 bacon and sausages
>     391,211 bread products (wholemeal light breads, ciabattas, bagels, crusty rolls, garlic breads, soft rolls, sausage rolls, stoneground wholemeal breads)
>     39,194 slimming products
>     19,122 muffins
>     15,383 milkshake mixes
>
> N retained=32,110 active households reporting 36,071,406 product-level purchases of food and beverages

> **Correction** of nutritional information (1.4%), measurement unit (0.09%) or pack number (0.1%)
>
> Main foods corrected: Eggs, Baguettes, Naan breads, Pitta breads, Scones, Crumpets and pikelets, Thins, Buns
>
> Main beverages corrected: Ales (beer) and wine energy values
>
> Correction methods: 1) adjust reported nutritional information (per unit vs. per 100g); 2) use identical product information at a different transaction; 3) use McCance and Widdowson's composition of food integrated dataset if unsure

> **Aggregation** across purchases made by households during the year for each food and beverage group and **exclusion** if <14 days of purchases reported in any quarter (19,408 households and 16,853,219 purchases excluded)
>
> N retained=12,702 active households reporting 19,218,187 product-level purchases of food and beverages

> **Exclusion** of households with extreme values on any beverage type (>364 L per household member per year) or beverage purchases (>500L per household member per year): 207 households and 362,908 purchases excluded.
>
> N retained=12,495 active households reporting 18,855,279 product-level purchases of food and beverages

> **Exclusion** of households who purchased ≤ 52L of beverages per household member per year: 3,820 households and 4,848,053 purchases excluded.
>
> **Analysis sample:** 8,675 households reporting 14,007,226 product-level purchases of food and beverages in 2016 (52-week period starting in January 2016)

**Fig 1. Data flow chart.** Ethical approval was not required as the data were obtained in anonymised format. Upon joining the panel, participants agree to the terms and conditions of Great Britain Kantar Fast-Moving Consumer Goods (see https://www.kantarworldpanel.com/en for contact details).

professional occupations (C1); skilled manual workers (C2); semi- or unskilled manual workers (D); and state pensioners, casual or lowest grade workers, and those unemployed with state benefits (E) [32]. Household income was categorised as <£20,000, £20,000–29,999, £30,000–£39,999, £40,000–£49,999, and ≥£50,000. BMI was categorised as underweight, normal weight, overweight, and obese using WHO cutoffs [33].

Non-response in demographic data was relatively rare; income was missing for 15.1% of households; BMI was missing for 15.2% of respondents (main shoppers); and other items were fully observed. Missingness did not depend on the latent class variable (i.e., outcome); complete cases were therefore analysed ($n$ = 6,432 out of 8,675; 74%) [34].

## Statistical analyses

LCA is a data-driven method used to explore population heterogeneity. It does this by identifying subpopulations, called latent classes, that share similar item response patterns. By estimating probabilities of observed response conditional on class membership, LCA allows estimates of posterior probability for each household belonging to each latent class [35]. LCA is therefore a very useful tool for identifying potential targets for interventions rather than relying on simple targeting based on population characteristics such as income or weight status [21].

A series of LCA models specifying latent class counts from 1 to 10 were fitted. To decide on the number of latent classes, we assessed successive models to identify the model with a combination of the lowest Akaike information criterion (AIC) and Bayesian information criterion (BIC) values, low Vuong–Lo–Mendell–Rubin adjusted likelihood ratio test (VLMR-LRT) value [36], low bootstrap likelihood ratio test (BLRT) value [37,38], and high standardised entropy [35]. Ultimately, the decision on the number of latent classes was guided by substantive interpretability, preferring a model whose class separation was simplest to articulate [39], and for which the relative size of classes was no lower than approximately 5% of the sample [40].

A 3-step approach was used to relate latent classes to demographic covariates, wherein Step 1 was to fit a LCA model, Step 2 was to assign households to latent classes, and Step 3 was to relate covariates to assigned latent classes using multinomial logistic regression [41].

To investigate the informative value of beverage classification, multivariable models predicting energy and nutrients purchased from food and beverages were developed, using a stepwise approach. Due to classification errors in the results from the DU3STEP method, we used the modified Bolck–Croon–Hagenaars (BCH) approach, as recommended [42]. Models were adjusted for household size and number of children.

Data were prepared using Stata MP version 15, and analyses conducted using Mplus version 8.2. All analyses were conducted in 2019–2020.

## Sensitivity analyses

Sensitivity analyses were conducted to ensure that the exclusion criteria and the categorisation of the latent class indicators did not affect the results. First, we broadened our definition of outliers by excluding households that, for any of the 6 latent class indicators, purchased volume in greater quantity than 3 SD (final $N$ = 7,446). Second, we used quintiles instead of tertiles of the beverage volumes as input in the LCA.

## Results

Table 1 reports characteristics of the 8,675 households that reported regular beverage purchases. Median volume of beverages purchased was 15.6 l (per household member/year) for SSBs, 43.2 l for diet beverages, 6.4 l for fruit juices and milk-based beverages, 5.6 l for beer and

**Table 1. Selected characteristics of the 8,675 British households that reported regular take-home beverage purchases between 4 January 2016 and 1 January 2017 (52 weeks).**

| Characteristic | Value | N missing (%) |
|---|---|---|
| **Household characteristics** | | |
| Region, *N* (%) | | |
| London | 1,214 (14.0%) | |
| Midland | 1,228 (14.2%) | |
| North east | 470 (5.4%) | |
| Yorkshire | 1,126 (13.0%) | |
| Lancashire | 967 (11.2%) | |
| South | 939 (10.8%) | |
| Scotland | 893 (10.3%) | |
| East England | 781 (9.0%) | |
| Wales and West | 757 (8.7%) | |
| South West | 300 (3.5%) | |
| Annual household income, *N* (%) | | 1,310 (15.1%) |
| <£20,000 | 2,508 (34.0%) | |
| £20,000–29,999 | 1,700 (23.1%) | |
| £30,000–39,999 | 1,219 (16.6%) | |
| £40,000–49,999 | 769 (10.4%) | |
| ≥£50,000 | 1,169 (15.9%) | |
| Life stage, *N* (%) | | |
| Pre-family | 649 (7.5%) | |
| Young family | 636 (8.3%) | |
| Middle family | 733 (8.5%) | |
| Older family | 822 (9.5%) | |
| Older dependents | 1,267 (14.6%) | |
| Empty nesters | 2,374 (27.4%) | |
| Retired | 2,194 (25.3%) | |
| Occupational social grade of the main shopper[a], *N* (%) | | |
| A&B | 1,689 (19.5%) | |
| C1 | 3,442 (39.7%) | |
| C2 | 1,574 (18.2%) | |
| D | 1,226 (14.1%) | |
| E | 744 (8.6%) | |
| BMI category of the main shopper, *N* (%) | | 1,315 (15.2%) |
| Underweight | 111 (1.3%) | |
| Normal | 2,572 (35.0%) | |
| Overweight | 2,584 (35.1%) | |
| Obese | 2,093 (28.4%) | |
| **Beverages purchased (litres/household member/year), median (minimum–maximum)** | | |
| SSBs | | |
| Low tertile | 3.4 (0.0–8.5) | |
| Middle tertile | 15.6 (8.5–27.3) | |
| High tertile | 48.4 (27.3–354.2) | |
| All | 15.6 (0.0–354.2) | |
| Diet beverages | | |

(*Continued*)

**Table 1.** (Continued)

| Characteristic | Value | N missing (%) |
|---|---|---|
| Low tertile | 7.6 (0.0–23.8) | |
| Middle tertile | 43.2 (23.9–69.6) | |
| High tertile | 111.0 (69.7–363.3) | |
| All | 43.2 (0.0–363.3) | |
| Fruit juices/milk-based beverages | | |
| Low tertile | 0.7 (0.0–2.7) | |
| Middle tertile | 6.4 (2.7–13.0) | |
| High tertile | 26.3 (13.0–183.1) | |
| All | 6.4 (0.0–183.1) | |
| Beer and cider | | |
| Low tertile | 0.0 (0.0–2.0) | |
| Middle tertile | 5.6 (2.0–12.8) | |
| High tertile | 31.9 (12.8–355.0) | |
| All | 5.6 (0.0–355.0) | |
| Wine | | |
| Low tertile | 0.3 (0.0–1.3) | |
| Middle tertile | 3.6 (1.3–9.0) | |
| High tertile | 23.8 (9.0–343.1) | |
| All | 3.5 (0.0–343.1) | |
| Bottled water | | |
| Low tertile | 0.0 (0.0–0.4) | |
| Middle tertile | 2.3 (0.4–8.4) | |
| High tertile | 31.5 (8.5–360.5) | |
| All | 2.3 (0.0–360.5) | |
| **Energy purchased (kcal/household member/day), mean (SD)** | | |
| Beverages[b] | 95.4 (82.8) | |
| Beverages[b] excluding alcohol | 46.3 (43.2) | |
| All food and beverages[c] | 1,798.2 (657.2) | |
| Percent from beverages | 5.6 (4.5) | |
| Percent from beverages excluding alcohol | 2.7 (2.4) | |
| Percent from sweet snacks (chocolates, confectionery, puddings, biscuits) | 16.3 (7.1) | |
| Percent from fruits and vegetables | 7.6 (4.0) | |
| Percent from less healthy food and beverages[d] | 49.9 (8.7) | |
| **Nutrients purchased (grams/1,000 kcal), mean (SD)** | | |
| Fat | 39.9 (5.3) | |
| Saturated fat | 15.4 (2.7) | |
| Protein | 34.1 (5.7) | |
| Carbohydrates | 115.3 (14.8) | |
| Sugar | 55.3 (12.3) | |
| NSP fibre | 9.0 (2.1) | |

(*Continued*)

**Table 1.** (Continued)

| Characteristic | Value | N missing (%) |
|---|---|---|
| Sodium | 1.0 (0.2) | |

Data are from Great Britain Kantar Fast-Moving Consumer Goods.

[a]A&B—higher and intermediate managerial, administrative, or professional occupations; C1—supervisory, clerical, and junior managerial administrative or professional occupations; C2—skilled manual workers; D—semi- or unskilled manual workers; E—state pensioners, casual or lowest grade workers, and those unemployed with state benefits.

[b]From the above categories.

[c]Value underreported by around 3% on average. In addition, some puddings, biscuits, and bread products, as well as all bacon and sausages, slimming products, and milkshake mixes, were excluded because of inconsistent nutrient information reported at the product level. Products excluded could account for up 130 kcal per household member per day.

[d]Defined using the nutrient profiling model of the UK Department of Health and Social Care.

NSP, non-starch polysaccharide; SSB, sugar-sweetened beverage.

cider, 3.5 l for wine, and 2.3 l for bottled water (Table 1). These beverages accounted for an average of 95.4 kcal, or 5.6% of total energy purchased per household member per day. In comparison, 16.3% of the energy purchased was obtained from sweet snacks, and 49.9% from less healthy food and beverages overall.

## Latent classes

Differing selection criteria favoured differing numbers of latent classes. BIC scores were lowest for the 7-class model. Entropy fluctuated between 0.52 and 0.71; the BLRT had $p < 0.001$ for all models; and VLMR-LRT did not improve after the 9-class model ($p = 1.000$). All models except for the 8- and 10-class models were reasonably balanced regarding count of members ($\geq$4%). S3 Appendix presents all fit statistics for models with 1 to 10 latent classes.

A 7-class model represented the optimal balance of model fit and interpretability (Fig 2; S4 Appendix). These 7 latent classes are interpreted in Table 2. Households purchasing higher volumes of SSBs were classified in 2 classes depending on their patterns of consumption of other beverages: some purchased a diversity of beverages (class 7, 'Diverse' class, 30%; median SSB volume = 22.4 l/household member/year); others purchased higher volumes of SSBs (median = 49.4 l) and diet beverages (median = 38.1 l; class 1, 'SSB', 18%). Households purchasing high volumes of diet beverages (median = 94.4 l) but very little other beverages were classified in class 2, 'Diet' (16%). High diet beverage volumes were also observed in classes 4 and 5 (median = 55.6 l and 34.6 l, respectively), which were also characterised by high volumes of beer and cider (median = 36.3 l; class 4, 'Beer & Cider', 7%) and wine (median = 25.5 l; class 5, 'Wine', 18%), respectively. The remaining 2 latent classes characterised households that purchased either a high volume of fruit juices and milk-based beverages (median = 30.0 l; class 3, 'Fruit & Milk', 6%) or a high volume of bottled water (median = 46.9 l; class 6, 'Water', 4%).

Table 2 also shows the average energy obtained from beverages, conditional on assigned class membership. Members of class Wine and class Diverse obtained the most calories from beverages (mean 149.9 kcal/household member/day, 95% CI 142.4–157.5, and 128.5, 95% CI 123.1–133.9, respectively), and members of class SSB obtained the most calories from non-alcoholic beverages (mean 92.6 kcal/household member/day, 95% CI 89.3–95.8). Members of class Diet and class Water obtained the lowest calories from all beverages combined.

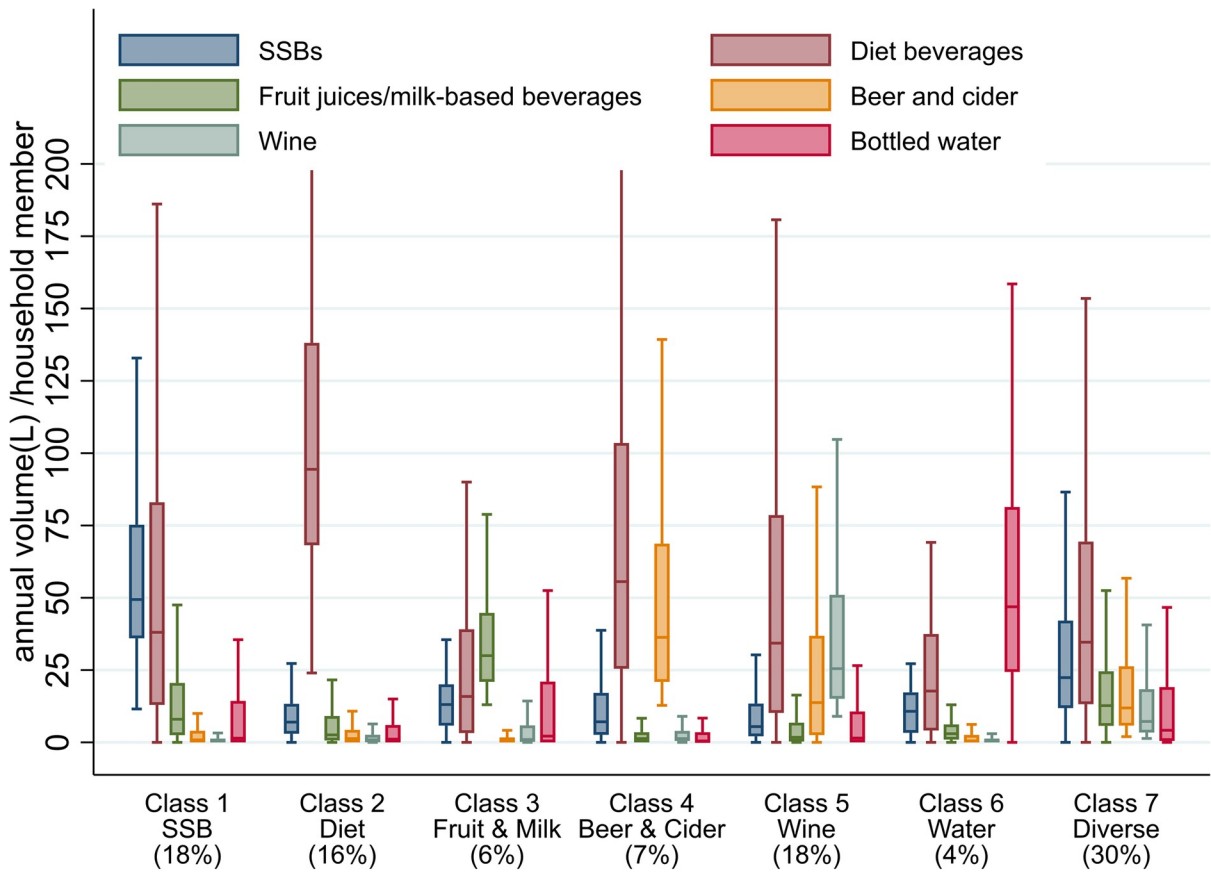

**Fig 2. Box-plot of the beverage categories by latent class (*N* = 8,675).** Data are from Great Britain Kantar Fast-Moving Consumer Goods. Results based on most likely class membership. SSB, sugar-sweetened beverage.

## Associations with socio-demographic variables and weight status

Table 3 shows the relative risk ratios (RRRs) of class membership as compared with the Diverse class in the final multivariable model. Region, income, life stage, and occupational social grade were all strongly predictive of latent class membership. Households living in London as opposed to other regions had greater probability of being in the SSB, Fruit & Milk, or Water class relative to being in the Diverse class (e.g., for Water: RRR = 0.26, 95% CI 0.10–0.70, *p* = 0.007, for South West compared to London). Living outside London (except Scotland, South, and South West) increased the probability of being in the Beer & Cider class relative to being in the Diverse class (RRRs ranged from 1.95, 95% CI 0.94–4.07, *p* = 0.073, to 3.43, 95% CI 1.50–7.82, *p* = 0.003). Higher levels of income were associated with decreased probability of being in the SSB/Fruit & Milk/Diet/Water classes as compared to the Diverse class (comparing household income ≥£50,000 to <£20,000, RRRs ranged from 0.19, 95% CI 0.12–0.30, *p* < 0.001, to 0.51, 95% CI 0.34–0.76, *p* = 0.001, in these classes).

Young families had a greater probability of being in the SSB class (RRR = 1.97, 95% CI 1.16–3.33, *p* = 0.012), whereas empty nesters were more likely to be in the Wine class, relative to being in the Diverse class (RRR = 4.94, 95% CI 2.40–10.19, *p* < 0.001). Retired households were more likely to be in the Fruit & Milk/Wine classes, relative to being in the Diverse class (RRR = 3.25, 95% CI 1.57–6.71, *p* = 0.001, and 6.11, 95% CI 2.90–12.89, *p* < 0.001, respectively).

**Table 2. Description of the latent classes (N = 8,675).**

| Latent class | Label | Description | Number of households (%)[a] | Energy from beverages (kcal/household member/day), mean (95% CI)[b] | Energy from beverages excluding alcohol (kcal/household member/day), mean (95% CI)[b] | Percent with overweight/ obesity (95% CI)[b,c] | Percent with obesity (95% CI)[b,c] |
|---|---|---|---|---|---|---|---|
| 1 | SSB | Households that purchased higher volumes of SSBs, but lower volumes of alcohol | 1,543 (18%) | 98.7 (95.2, 102.3) | 92.6 (89.3, 95.8) | 66.8 (63.7, 69.9) | 30.6 (27.5, 33.7) |
| 2 | Diet | Households that purchased high volumes of diet beverages but little of caloric beverages such as SSBs and alcohol | 1,402 (16%) | 17.9 (16.2, 19.7) | 15.9 (14.4, 17.4) | 72.5 (69.2, 75.8) | 41.2 (37.7, 44.7) |
| 3 | Fruit & Milk | Households that purchased high volumes of 100% fruit juices and milk-based beverages, but lower volumes of diet beverages and alcohol | 529 (6%) | 85.6 (78.2, 93.0) | 70.0 (66.3, 73.7) | 56.7 (50.6, 62.8) | 23.0 (17.7, 28.3) |
| 4 | Beer & Cider | Households that purchased high volumes of beer and cider, but little wine, SSBs, water, or juices | 644 (7%) | 70.4 (64.5, 76.4) | 8.6 (5.5, 11.7) | 62.5 (57.2, 67.8) | 31.0 (25.9, 36.1) |
| 5 | Wine | Households that purchased high volumes of wine, and alcohol, but little of SSBs or juices | 1,586 (18%) | 149.9 (142.4, 157.5) | 7.4 (5.2, 9.6) | 58.9 (55.4, 62.4) | 21.0 (17.9, 24.1) |
| 6 | Water | Households that purchased high volumes of bottled water and lower volumes of caloric beverages | 352 (4%) | 20.6 (15.8, 25.3) | 14.6 (12.8, 16.4) | 60.9 (53.5, 68.3) | 27.0 (20.1, 33.9) |
| 7 | Diverse | Households that purchased a diversity of beverages, including higher than average alcohol volumes | 2,619 (30%) | 128.5 (123.1, 133.9) | 66.4 (63.8, 69.1) | 62.0 (58.7, 65.3) | 25.6 (22.7, 28.5) |

Data are from Great Britain Kantar Fast-Moving Consumer Goods.

[a]Based on most likely class membership.

[b]Estimated using the manual 3-tep Bolck–Croon–Hagenaars method.

[c]Based on complete cases (N = 7,360).

SSB, sugar-sweetened beverage.

Households in the lowest occupational grade (grade E) were more likely to be in the SSB/Diet/Beer & Cider classes, relative to being in the Diverse class (e.g., for SSB: RRR = 9.27, 95% CI 3.47–24.80, $p < 0.001$, compared to grade A&B). Occupational differences showed a clear social gradient for classes SSB and Beer & Cider.

## Associations with weight status

Table 2 shows that classes Diet and SSB have the highest proportion of respondents with overweight or obesity (i.e., BMI > 25 kg/m$^2$) (72.5%, 95% CI 69.2%–75.8%, and 66.8%, 95% CI 63.7%–69.9%, respectively). The proportion with obesity was highest in class Diet (41.2%, 95% CI 69.2%–75.8%) and lowest in classes Wine (21.0%, 95% CI 17.9%–24.1%) and Fruit & Milk (23.0%, 95% CI 17.7%–28.3%).

These results are confirmed in multivariable analysis (Table 3). Respondents with obesity had higher probability of being in the Diet class, relative to being in the Diverse class (RRR = 1.99, 95% CI 1.50–2.65, $p < 0.001$), and respondents with obesity had lower probability of being in the Wine class, relative to being in the Diverse class (RRR = 0.60, 95% CI 0.42–0.85, $p = 0.005$).

**Table 3. Relative risk ratio (95% CI) of membership in specified class compared to Diverse class in multinomial logistic regression (N = 6,432).**

| Characteristic | Class 1: SSB | p-Value | Class 2: Diet | p-Value | Class 3: Fruit & Milk | p-Value | Class 4: Beer & Cider | p-Value | Class 5: Wine | p-Value | Class 6: Water | p-Value |
|---|---|---|---|---|---|---|---|---|---|---|---|---|
| Region | | | | | | | | | | | | |
| London | Ref. | | Ref. | | Ref. | | Ref. | | Ref. | | Ref. | |
| Midland | 0.98 (0.65, 1.48) | 0.923 | 1.23 (0.77, 1.95) | 0.392 | 0.44 (0.25, 0.79) | 0.006 | 1.95 (0.94, 4.07) | 0.073 | 1.14 (0.67, 1.96) | 0.631 | 0.30 (0.15, 0.59) | 0.001 |
| North East | 0.76 (0.42, 1.35) | 0.346 | 1.44 (0.79, 2.62) | 0.231 | 0.31 (0.11, 0.89) | 0.029 | 3.43 (1.50, 7.82) | 0.003 | 1.95 (1.01, 3.75) | 0.046 | 0.09 (0.01, 0.84) | 0.034 |
| Yorkshire | 0.66 (0.43, 1.02) | 0.062 | 1.36 (0.86, 2.15) | 0.190 | 0.33 (0.18, 0.61) | <0.001 | 2.31 (1.14, 4.66) | 0.020 | 1.09 (0.65, 1.85) | 0.740 | 0.22 (0.10, 0.47) | <0.001 |
| Lancashire | 0.66 (0.40, 1.06) | 0.087 | 1.61 (0.99, 2.62) | 0.054 | 0.39 (0.2, 0.76) | 0.006 | 2.84 (1.35, 5.99) | 0.006 | 1.84 (1.07, 3.17) | 0.028 | 0.49 (0.26, 0.95) | 0.035 |
| South | 0.79 (0.50, 1.22) | 0.285 | 1.27 (0.79, 2.04) | 0.324 | 0.37 (0.2, 0.68) | 0.001 | 0.92 (0.38, 2.22) | 0.858 | 0.97 (0.56, 1.70) | 0.922 | 0.41 (0.22, 0.78) | 0.007 |
| Scotland | 0.89 (0.57, 1.41) | 0.631 | 1.38 (0.85, 2.25) | 0.193 | 0.55 (0.31, 0.97) | 0.039 | 1.04 (0.44, 2.48) | 0.932 | 1.16 (0.67, 2.02) | 0.595 | 0.20 (0.07, 0.54) | 0.002 |
| East England | 0.85 (0.53, 1.37) | 0.497 | 1.41 (0.85, 2.36) | 0.184 | 0.68 (0.37, 1.26) | 0.215 | 2.37 (1.10, 5.10) | 0.028 | 1.18 (0.65, 2.13) | 0.582 | 0.35 (0.16, 0.79) | 0.012 |
| Wales and West | 0.61 (0.36, 1.05) | 0.072 | 1.36 (0.79, 2.33) | 0.268 | 0.05 (0.25, 1.00) | 0.049 | 2.32 (1.05, 5.13) | 0.038 | 1.70 (0.93, 3.13) | 0.086 | 0.48 (0.23, 1.01) | 0.054 |
| South West | 0.42 (0.22, 0.80) | 0.008 | 0.76 (0.40, 1.45) | 0.412 | —[a] | —[a] | 0.96 (0.35, 2.65) | 0.942 | 0.87 (0.41, 1.82) | 0.707 | 0.26 (0.10, 0.70) | 0.007 |
| Annual income | | | | | | | | | | | | |
| <£20,000 | Ref. | | Ref. | | Ref. | | Ref. | | Ref. | | Ref. | |
| £20,000–29,999 | 0.75 (0.54, 1.04) | 0.088 | 1.03 (0.73, 1.44) | 0.887 | 0.70 (0.43, 1.12) | 0.136 | 1.12 (0.72, 1.75) | 0.602 | 1.22 (0.83, 1.80) | 0.318 | 0.86 (0.49, 1.50) | 0.593 |
| £30,000–39,999 | 0.53 (0.37, 0.76) | 0.001 | 0.73 (0.50, 1.05) | 0.092 | 0.50 (0.30, 0.84) | 0.009 | 0.68 (0.40, 1.16) | 0.154 | 1.03 (0.67, 1.57) | 0.904 | 0.63 (0.34, 1.15) | 0.133 |
| £40,000–49,999 | 0.33 (0.21, 0.50) | <0.001 | 0.57 (0.38, 0.87) | 0.010 | 0.31 (0.16, 0.62) | 0.001 | 0.53 (0.27, 1.02) | 0.059 | 0.61 (0.35, 1.05) | 0.073 | 0.48 (0.24, 0.98) | 0.044 |
| ≥£50,000 | 0.19 (0.12, 0.30) | <0.001 | 0.51 (0.34, 0.76) | 0.001 | 0.43 (0.25, 0.75) | 0.003 | 0.69 (0.39, 1.24) | 0.218 | 0.81 (0.50, 1.30) | 0.379 | 0.28 (0.14, 0.56) | <0.001 |
| Life stage | | | | | | | | | | | | |
| Pre-family | Ref. | | Ref. | | Ref. | | Ref. | | Ref. | | Ref. | |
| Young family | 1.97 (1.16, 3.33) | 0.012 | 1.46 (0.88, 2.44) | 0.142 | 1.27 (0.51, 3.15) | 0.612 | 1.57 (0.70, 3.53) | 0.272 | 0.23 (0.02, 2.22) | 0.203 | 1.53 (0.67, 3.47) | 0.309 |
| Middle family | 1.16 (0.68, 2.00) | 0.586 | 1.33 (0.81, 2.19) | 0.258 | 1.74 (0.76, 4.00) | 0.192 | 1.18 (0.52, 2.69) | 0.687 | 0.98 (0.37, 2.58) | 0.960 | 1.22 (0.53, 2.79) | 0.639 |
| Older family | 1.18 (0.73, 1.91) | 0.503 | 0.74 (0.46, 1.20) | 0.220 | 1.23 (0.56, 2.73) | 0.610 | 0.42 (0.16, 1.10) | 0.077 | 0.08 (0.00, 60.38) | 0.258 | 0.46 (0.18, 1.20) | 0.114 |
| Older dependents | 1.40 (0.86, 2.27) | 0.176 | 0.93 (0.58, 1.49) | 0.772 | 1.62 (0.70, 3.74) | 0.256 | 1.19 (0.56, 2.53) | 0.646 | 1.89 (0.83, 4.27) | 0.128 | 1.10 (0.50, 2.40) | 0.818 |
| Empty nesters | 1.14 (0.73, 1.77) | 0.566 | 0.98 (0.65, 1.50) | 0.939 | 1.32 (0.63, 2.78) | 0.467 | 1.90 (0.98, 3.69) | 0.058 | 4.94 (2.40, 10.19) | <0.001 | 0.89 (0.43, 1.83) | 0.750 |
| Retired | 0.91 (0.56, 1.46) | 0.693 | 0.91 (0.58, 1.44) | 0.693 | 3.25 (1.57, 6.71) | 0.001 | 1.51 (0.75, 3.05) | 0.250 | 6.11 (2.90, 12.89) | <0.001 | 1.16 (0.55, 2.44) | 0.698 |
| Occupational social grade[b] | | | | | | | | | | | | |
| A&B | Ref. | | Ref. | | Ref. | | Ref. | | Ref. | | Ref. | |
| C1 | 1.21 (0.84, 1.76) | 0.307 | 0.85 (0.62, 1.17) | 0.313 | 0.68 (0.44, 1.06) | 0.086 | 1.33 (0.74, 2.38) | 0.345 | 0.77 (0.53, 1.11) | 0.155 | 0.44 (0.27, 0.73) | 0.002 |
| C2 | 1.31 (0.86, 2.01) | 0.207 | 0.89 (0.61, 1.30) | 0.555 | 0.65 (0.37, 1.12) | 0.121 | 2.84 (1.55, 5.21) | 0.001 | 0.68 (0.43, 1.06) | 0.090 | 0.76 (0.42, 1.37) | 0.354 |

*(Continued)*

**Table 3.** (Continued)

| Characteristic | Class 1: SSB | p-Value | Class 2: Diet | p-Value | Class 3: Fruit & Milk | p-Value | Class 4: Beer & Cider | p-Value | Class 5: Wine | p-Value | Class 6: Water | p-Value |
|---|---|---|---|---|---|---|---|---|---|---|---|---|
| D | 1.76 (1.12, 2.77) | 0.014 | 0.91 (0.58, 1.41) | 0.658 | 0.35 (0.17, 0.71) | 0.004 | 2.67 (1.35, 5.29) | 0.005 | 0.67 (0.40, 1.11) | 0.120 | 0.58 (0.29, 1.17) | 0.128 |
| E | 9.27 (3.47, 24.80) | <0.001 | 5.16 (1.96, 13.55) | 0.001 | 1.23 (0.37, 4.03) | 0.738 | 6.24 (1.92, 20.27) | 0.002 | 1.70 (0.57, 5.04) | 0.337 | 1.74 (0.44, 6.81) | 0.428 |
| BMI category | | | | | | | | | | | | |
| Underweight | 3.61 (0.84, 15.54) | 0.084 | 1.64 (0.32, 8.40) | 0.551 | 0.50 (0.02, 13.32) | 0.681 | 6.44 (1.11, 37.34) | 0.038 | 2.09 (0.32, 13.61) | 0.441 | 1.22 (0.10, 14.55) | 0.876 |
| Normal | Ref. | | Ref. | | Ref. | | Ref. | | Ref. | | Ref. | |
| Overweight | 1.03 (0.79, 1.35) | 0.823 | 1.07 (0.81, 1.43) | 0.620 | 0.70 (0.48, 1.02) | 0.061 | 0.84 (0.56, 1.25) | 0.390 | 0.79 (0.57, 1.07) | 0.129 | 0.77 (0.49, 1.21) | 0.259 |
| Obese | 1.12 (0.83, 1.49) | 0.462 | 1.99 (1.50, 2.65) | <0.001 | 0.70 (0.45, 1.07) | 0.102 | 1.12 (0.74, 1.70) | 0.602 | 0.60 (0.42, 0.85) | 0.005 | 0.92 (0.57, 1.49) | 0.748 |

Data are from Great Britain Kantar Fast-Moving Consumer Goods.

[a]Parameter could not be estimated due to small $N$.

[b]A&B—higher and intermediate managerial, administrative, or professional occupations; C1—supervisory, clerical, and junior managerial administrative or professional occupations; C2—skilled manual workers; D—semi- or unskilled manual workers; E—state pensioners, casual or lowest grade workers, and those unemployed with state benefits.

SSB, sugar-sweetened beverage.

## Associations with other food and beverage purchases

Beverage classes were also predictive of the energy and nutrients of food purchased by the households (Table 4). Households in classes SSB and Diverse purchased more energy relative to other classes (1,943.6 kcal/household member/day, 95% CI 1,901.7–1,985.6, and 1,933.6, 95% CI 1,891.0–1,976.3, respectively). Households in classes SSB and Diet obtained a greater share of energy from sweet snacks (18.5%, 95% CI 18.1%–19.0%, and 18.8%, 95% CI 18.3%–19.3%, respectively) compared to other classes. Furthermore, households in class SSB had both the lowest share of energy from fruits and vegetables (6.0%, 95% CI 5.8%–6.3) and the highest share of energy from less healthy food and beverages (54.6%, 95% CI 54.0%–55.1%). Households in class Diet had the second highest share of energy from less healthy food and beverages (51.2%, 95% CI 50.6%–51.7%). Conversely, households in class Wine obtained the lowest proportion of energy from less healthy products (41.6%, 95% CI 38.1%–45.0%), including from sweet snacks (11.2%, 95% CI 9.3%–13.1%).

Fat, saturated fat, and sodium content of purchases varied little across latent classes (Table 4). Households in classes SSB and Fruit & Milk obtained the highest share of their energy from sugar (62.9 g/1,000 kcal, 95% CI 62.1–63.6, and 63.0 g/1,000 kcal, 95% CI 61.7–64.4, respectively). The protein content of purchases was lowest for households in class SSB (32.3 g/1,000 kcal, 95% CI 31.9–32.6) and highest for households in class Wine (36.0 g/1,000 kcal, 95% CI 34.7–37.3). Conversely, households in class Wine obtained the smallest share of their energy from saturated fat (14.1 g/1,000 kcal, 95% CI 13.1–15.1) and sugar (41.8 g/1,000 kcal, 95% CI 37.0–46.6), relative to other classes.

## Sensitivity analysis

Analyses using the restricted sample ($N = 7,446$) and the quintile beverage indicators did not alter the main results (S5 Appendix). The restricted sample indicated that both the 6-class and

**Table 4. Mean (95% CI) energy and nutrient content of purchases by latent class (N = 8,675).**

| Variable | Class 1: SSB | Class 2: Diet | Class 3: Fruit & Milk | Class 4: Beer & Cider | Class 5: Wine | Class 6: Water | Class 7: Diverse |
|---|---|---|---|---|---|---|---|
| **Energy** | | | | | | | |
| Total (kcal/household member/day)[a] | 1,943.6 (1,901.7, 1,985.6) | 1,731.3 (1,690.2, 1,772.3) | 1,692.5 (1,619.8, 1,765.2) | 1,729.9 (1,667.1, 1,792.7) | 1,458.5 (1,158.6, 1,758.5) | 1,562.4 (1,475.8, 1,648.9) | 1,933.6 (1,891.0, 1,976.3) |
| Pecent from sweet snacks | 18.5 (18.1, 19.0) | 18.8 (18.3, 19.3) | 16.7 (15.8, 17.5) | 15.1 (14.4, 15.8) | 11.0 (8.8, 13.3) | 16.7 (15.6, 17.7) | 15.3 (14.9, 15.8) |
| Pecent from fruits and vegetables | 6.0 (5.8, 6.3) | 7.9 (7.6, 8.2) | 8.8 (8.3, 9.3) | 7.4 (6.9, 7.8) | 8.4 (7.9, 9) | 9.4 (8.7, 10.1) | 7.4 (7.1, 7.6) |
| Pecent from beverages, excluding alcohol | 5.3 (5.1, 5.4) | 1.0 (0.9, 1.1) | 4.4 (4.2, 4.6) | 0.6 (0.4, 0.7) | 0.0 (-0.6, 0.3) | 1.2 (1.0, 1.3) | 3.7 (3.6, 3.9) |
| Pecent from beverages | 5.6 (5.4, 5.8) | 1.2 (1.1, 1.3) | 5.3 (4.9, 5.7) | 4.6 4.2, 5.0) | 10.2 (7.8, 12.6) | 1.6 (1.3, 1.9) | 7.1 (6.8, 7.4) |
| Pecent from less healthy food and beverages[b] | 54.6 (54.0, 55.1) | 51.2 (50.6, 51.7) | 48.0 (47.0, 49.0) | 48.1 (47.2, 49.1) | 41.6 (38.1, 45.0) | 49.2 (47.9, 50.5) | 50.4 (49.9, 51.0) |
| **Nutrients (grams/1,000 kcal)** | | | | | | | |
| Fat | 39.8 (39.5, 40.1) | 40.0 (39.6, 40.4) | 39.8 (39.1, 40.4) | 39.4 (38.8, 40.0) | 37.9 (36.2, 39.6) | 41.6 (40.7, 42.4) | 40.2 (39.9, 40.6) |
| Saturated fat | 15.5 (15.3, 15.6) | 15.6 (15.4, 15.8) | 15.7 (15.4, 16.0) | 14.9 (14.6, 15.2) | 14.1 (13.1, 15.1) | 15.8 (15.4, 16.3) | 15.6 (15.5, 15.8) |
| Protein | 32.3 (31.9, 32.6) | 35.7 (35.3, 36.1) | 33.8 (33, 34.6) | 35.7 (35, 36.3) | 36.0 (34.7, 37.3) | 35.1 (34.3, 35.9) | 33.2 (32.8, 33.5) |
| Carbohydrates | 123.9 (123.1, 124.7) | 119.2 (118.3, 120.1) | 121.3 (119.6, 122.9) | 112.0 (110.5, 113.4) | 99.1 (94.2, 104.1) | 117.2 (115.3, 119.1) | 114.0 (113.1, 114.9) |
| Sugar | 62.9 (62.1, 63.6) | 54.0 (53.1, 54.8) | 63.0 (61.7, 64.4) | 49.4 (48.1, 50.6) | 41.8 (37.0, 46.6) | 54.8 (53.2, 56.4) | 56.1 (55.4, 56.9) |
| NSP fibre | 8.2 (8.1, 8.3) | 9.5 (9.3, 9.6) | 9.6 (9.4, 9.9) | 9.0 (8.7, 9.2) | 9.2 (8.9, 9.4) | 10.0 (9.7, 10.4) | 8.7 (8.6, 8.8) |
| Sodium | 1.0 (1.0, 1.0) | 1.1 (1.1, 1.1) | 0.9 (0.9, 1.0) | 1.1 (1.1, 1.1) | 1.0 (1.0, 1.1) | 1.0 (1.0, 1.0) | 1.0 (1.0, 1.0) |

Data are from Great Britain Kantar Fast-Moving Consumer Goods. Each energy/nutrient variable was treated as an auxiliary variable, and means (95% CIs) were estimated using the manual 3-Step Bolck–Croon–Hagenaars method, adjusting for household size and number of children to account for unequal purchases due to household composition (results displayed for sample means: household size of 2.5 and a number of children of 0.42).

[a]Value underreported by around 3% on average. In addition, some puddings, biscuits, and bread products, as well as all bacon and sausages, slimming products, and milkshake mixes, were excluded because of inconsistent nutrient information reported at the product level. Products excluded could account for up 130 kcal per household member per day.

[b]Defined using the UK Department of Health and Social Care nutrient profiling model.

NSP, non-starch polysaccharide; SSB, sugar-sweetened beverage.

the 7-class models had good fit. The 6-class model essentially grouped together households with high alcohol purchases in 1 latent class, instead of splitting them between Wine and Beer & Cider classes. The quintile analysis identified the same 6-class model as best fitting. Other latent classes had similar interpretation, and associations with socio-demographic variables and food purchase variables were of similar magnitude.

## Discussion

This LCA identified 7 different types of households on the basis of regular beverage purchasing behaviour: SSB, Diet, Fruit & Milk, Beer & Cider, Wine, Water, and Diverse. The Diverse class had greatest representation in the sample (30%), followed by the SSB (18%), Wine (18%), and Diet (16%) classes. Income was positively associated with being classified in the Diverse class, whereas low social grade was more likely for households classified in the classes purchasing high volumes of SSBs, diet beverages, and beer and cider. Overweight/obesity was above average in the classes SSB and Diet. Obesity was more likely in the class Diet despite households obtaining less energy from beverages in that class, relative to other classes. When looking at total food and beverage purchases, households from the class SSB obtained higher total energy, a smaller proportion of energy from fruits and vegetables, and a greater proportion of energy

from less healthy food and beverages than other classes. A greater proportion of energy from less healthy products, including from sweet snacks, was also purchased by households in the Diet class.

Beverage purchasing is strongly correlated with the purchase of other foods [19,20]. As reported in young people in the US [43], households purchasing SSBs tend to have worse diets compared to others, purchasing more energy in greater proportion from less healthy foods, such as sweet snacks. We also found that households purchasing more diet beverages also purchased a greater share of their energy from sweet snacks, indicating possible compensatory behaviours. Conversely, households regularly purchasing wine but avoiding SSBs (class 5) had healthier shopping baskets compared to other groups. In accordance with previous studies, we found that low income and lower occupational social grade were associated with purchases of SSBs and diet beverages [9]. We found evidence that the class with high SSB purchases was more likely to contain households where the youngest child was 0–4 years old. We did not find such evidence for households with older children or teenagers.

Previous research suggests that SSB consumption is associated with higher prevalence of overweight and obesity [5]. Our results indicate that though households with high SSB purchases have generally worse dietary purchase behaviours, this is associated with higher BMI (i.e., >25 kg/m$^2$), but not with obesity. Conversely, households characterised by diet beverage purchases had higher prevalence of obesity. Results should nonetheless be interpreted cautiously as this analysis was cross-sectional—changes in dietary behaviours (e.g., substitution of SSBs with diet beverages as a strategy to lose weight) might not be reflected in weight status. The lack of information on physical activity and out-of-home dietary behaviours (e.g., food eaten on the go or in restaurants) may partly explain some of these results.

This study complements modelling or price elasticity analyses that rely on pre-defined sub-populations of interest, such as high SSB consumers or those with low household income, to study the effect heterogeneity of policies. Unlike these analyses, we used LCA, a data-driven method that allowed us to identify patterns of food and beverage consumption that might allow better targeting of obesity policy. Our analysis informs policy in several ways. First, our results indicate that a policy such as the SDIL is well targeted, with 18% of the sample purchasing relatively high levels of SSBs (median approximately 1 l per week per household member) and a further 30% purchasing about 0.5 l per week per household member. Our findings also confirmed that those consuming higher levels of SSBs are more likely to have lower socio-economic status and therefore to respond to pricing policies.

Second, as diet beverages are the most purchased across beverage types, reformulation of SSBs to contain less sugar is likely to be accepted by consumers. However, full substitution patterns are more complex, and consideration needs to be given to broadening policies to cover fruit juices and milk-based beverages as those purchasing high or medium levels of SSBs (SSB and Diverse classes) are also the second and third largest purchasers of fruit juices and milk-based beverages.

Third, SSB purchases on their own may not represent the highest risk for obesity, as a stronger association with BMI was found for those purchasing diet beverages. When looking at overall food purchases, both the SSB and Diet classes had the worst purchase behaviours, characterised by a greater proportion of energy obtained from sweet snacks (approximately 18%). Therefore, our results suggest that effective policies need to broaden their targets beyond SSBs to reduce obesity, and policies such as the SDIL alone may miss some households at higher risk. This reinforces the argument that policies aimed at tackling obesity should be extended to sweet snacks, as a major source of excess energy amongst households at higher risk [27].

## Limitations

First, the main limitation of our analysis is, in common with other studies, that we were unable to account for out-of-home purchases, which account for 25%–39% of total food and beverage expenditures [31]. Patterns of beverage purchasing for the home might differ from those made outside of home. Second, there are potential limitations related to the type of data used, as participants might suffer from fatigue bias, with reporting becoming less accurate over time. Kantar monitors these potential biases by identifying and excluding problematic panellists, and we have further restricted the analysis to households consistently reporting purchases [24]. Third, BMI data were self-reported and only available for the main shopper in the household. Fourth, we were unable to take into account panel weights in the LCA, which implies that the latent class distribution of households might not be fully representative of the GB population. This also means that energy and nutrient values are underreported in this study, by about 3% [23]. However, this effect was minimised by the restriction of our analyses to households that regularly reported purchases.

A main strength of this study is the use of product-specific data on a large panel of the GB population. Other strengths include objective scanning of purchases, which avoids bias inherent in self-reported dietary intake [44,45].

## Conclusion

Our analyses of large-scale consumer panel data have allowed us to identify 7 types of households on the basis of regular beverage purchasing behaviour. Medium-to-high levels of SSBs were purchased by 48% of the sample. Households that mainly purchased high volumes of diet beverages were more likely to have obesity. Purchases of high volumes of either SSBs or diet beverages were indicative of relatively poorer diets, as households in these classes tended to have greater purchases of less healthy foods, obtaining approximately 18% of energy from sweet snacks. These results suggest that these households might additionally benefit from policies targeting unhealthy foods, such as sweet snacks, as a way of reducing excess energy intake. More research is needed to understand how beverage and food consumption patterns will be affected by policies that aim to reduce sugar purchases and consumption more broadly.

## Supporting information

**S1 Appendix. STROBE checklist.**
(DOC)

**S2 Appendix. Food and beverage group definitions.**
(DOCX)

**S3 Appendix. Model comparison of the latent class analysis.**
(DOCX)

**S4 Appendix. Classification of beverage purchasing into 7 latent classes using 8,675 British households that reported regular beverage purchases between 4 January 2016 and 1 January 2017 (52 weeks).**
(DOCX)

**S5 Appendix. Sensitivity analyses.**
(DOCX)

## Author Contributions

**Conceptualization:** Nicolas Berger, Steven Cummins, Richard D. Smith, Laura Cornelsen.

**Data curation:** Nicolas Berger, Alexander Allen, Laura Cornelsen.

**Formal analysis:** Nicolas Berger, Alexander Allen.

**Funding acquisition:** Laura Cornelsen.

**Investigation:** Laura Cornelsen.

**Methodology:** Nicolas Berger, Laura Cornelsen.

**Supervision:** Steven Cummins, Richard D. Smith, Laura Cornelsen.

**Validation:** Nicolas Berger.

**Writing – original draft:** Nicolas Berger, Alexander Allen.

**Writing – review & editing:** Nicolas Berger, Steven Cummins, Richard D. Smith, Laura Cornelsen.

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
