## [Editor Report · Decision Letter 0]

17 Feb 2020

Dear Dr Berger, 

Thank you for submitting your manuscript entitled "Patterns of beverage purchases amongst British households:a latent class analysis" for consideration by PLOS Medicine.

Your manuscript has now been evaluated by the PLOS Medicine editorial staff [as well as by an academic editor with relevant expertise] and I am writing to let you know that we would like to send your submission out for external peer review.

Kind regards,

Adya Misra, PhD,

Senior Editor

PLOS Medicine

---

## [Decision Letter · Decision Letter 1]

9 Apr 2020

Dear Dr. Berger,

Thank you very much for submitting your manuscript "Patterns of beverage purchases amongst British households:a latent class analysis" (PMEDICINE-D-20-00344R1) for consideration at PLOS Medicine. 

[LINK]

In light of these reviews, I am afraid that we will not be able to accept the manuscript for publication in the journal in its current form, but we would like to consider a revised version that addresses the reviewers' and editors' comments. Obviously we cannot make any decision about publication until we have seen the revised manuscript and your response, and we plan to seek re-review by one or more of the reviewers. 

We expect to receive your revised manuscript by Apr 30 2020 11:59PM. Please email us (plosmedicine@plos.org) if you have any questions or concerns.

We look forward to receiving your revised manuscript. 

Sincerely,

Adya Misra, PhD

Senior Editor 

PLOS Medicine

plosmedicine.org

Abstract

Methods and findings- please mention briefly the source of income and occupation social grade data. If this is available at FMCG please mention this. Please also mention the threshold of “regular purchases” and “high volumes of SSB”.

Conclusions

* Please address the study implications without overreaching what can be concluded from the data; the phrase "In this study, we observed ..." may be useful.

* Please interpret the study based on the results presented in the abstract, emphasizing what is new without overstating your conclusions.

* Please avoid vague statements such as "these results have major implications for policy/clinical care". Mention only specific implications substantiated by the results.

* Please avoid assertions of primacy ("We report for the first time....")

Methods

It might help to state which data were obtained from the FMCG early in the methods section

Is there an alternative phrase for “empty nesters” ? Not only colloquial, I do wonder if this type of label could be perceived as stigmatising

It would be helpful to define thresholds for beverage classes like “low wine” or “low diet” briefly. This information may be in the supplementary files but it would be beneficial in the main text for clarity. 

Please ensure that the study is reported according to the STROBE guideline, and include the completed STROBE checklist as Supporting Information. When completing the checklist, please use section and paragraph numbers, rather than page numbers. Please add the following statement, or similar, to the Methods: "This study is reported as per the Strengthening the Reporting of Observational Studies in Epidemiology STROBE guideline (S1 Checklist)."

Please report your study according to the relevant guideline, which can be found here: http://www.equator-network.org/

Results

Please provide 95% confidence intervals along with all p-values. In addition, please provide exact p-values unless p<0.001

Discussion

You note that BMI was self reported in the limitations section but I didn’t see this in the methods. Could you please provide this information there? 

Please use UK or Great Britain consistently in the manuscript

Please clarify if the FMCG is representative of the UK population. Please also clarify if the channel islands are included in the panel

Comments from the reviewers:

Reviewer #1: This is a statistical review of manuscript PMEDICINE-D-20-00344_R1. The manuscript reads well. I have two minor comments.

Minor comments:

* Abstract: you state ""… restricted our analyses to consumers who purchase beverages regularly (n=6,339)." I know that it is very clearly defined in the Methods, but perhaps define what you mean regularly in the Abstract for the reader who may only read the abstract. 

* Page 8: "complete cases were therefore analysed". How many were in the final analysis? I think that this is a crucial point to make. One would want to know how many (in absolute and relative terms) records out of the 6,339 have been used for the LCA. 

Reviewer #2: Thank you for the opportunity to review this interesting study. This study used a latent class analysis to characterise patterns of beverage purchases using data from the 2016 UK Kantar consumer panel. The authors identified 7 patterns with regard to beverage purchasing and related household and panel member characteristics to these patterns. I think the paper has implications for public policy, and the use of the consumer panel data is unique, although not without limitations (these are mostly discussed by the authors in the paper). My specific comments and suggestions are as follows:

1. You state that my "My study does not require an ethics statement." However, this study does involve human participants. In a recent study in PLOS Medicine (Taillie et al., 2020;17:e1003015) the authors have included a statement under methods that the study was exempt from review from their relevant review board as it uses secondary, de-identified data. Could you please check with your ethics committee to confirm that the current study does not need ethical review?

2. Pg 7. When you state that "For each household, we computed overall mean daily energy and nutrients purchased per household member from all food and beverages.", were adults and children treated equally, or was some allowance made for the fact that children in a household might have different energy/nutrient requirements and not eat as many of the purchased energy and nutrients?

3. Pg 7. When you refer to "selected food groups of public health concern", I found this wording ambiguous, I initially thought you meant you were interested in less healthy food and beverages which included the sub-groups of sweet snacks and fruit and vegetables. This is obviously not what you meant, so perhaps if you state there were 3 food groups of interest this might help. I suggest using the word "interest" instead of "concern" as concern has negative connotations and fruit and vegetables are 'healthy'. Also, can you please clarify whether 'less healthy' food and beverages and 'sweet snacks' are mutually exclusive food groups? Typically in nutrition papers food groups are mutually exclusive, and if there is some overlap here it would be useful to make that clear to the reader. 

4. I am interested in your rationale for mostly using % of purchased energy from food groups or nutrient density (e.g. in Table 5) although I note in Table 2 you present actual energy from purchased beverages. For studies that use dietary assessment of intakes the results are typically presented as % of energy or adjusted for energy intake to reduce measurement error (assuming that most participants misreport foods and beverages in the same direction and to a similar extent). However, this is not the case here as you are looking at purchases, and the consumer panel does not rely on self-reported intakes. On the other hand, I think it might help with the fact that the household composition differs, and houses have varying numbers of adults and children present.

5. Are the life stages named by Kantar? I assume they are but wonder if a couple who doesn't intend to have children would appreciate being labelled 'pre-family'. If they are labelled by Kantar I don't think you should change the names. Please make it clear in the text and perhaps in a footnote in Table 1 that the age range is (presumably) the age range of children in the household. What happens if a couple has two children aged 3 and 7? Would they be classed as young or middle family?

6. I note that you start with 32,110 households but after exclusions you end up with only 6,339 households. About 20,000 households appear to be excluded because they have fewer than 14 days of purchasing in any quarter. What is the rationale for applying this criteria? My concern is that you are making assumptions about the typical way for people to shop. To be included in your final sample, each household would have to do more than one shop a week, some households may only do a weekly shop or may even do a fort-nightly or monthly shop. I think you should only exclude households where the expenditure on food and beverages is implausible - I think that Kantar have already done this (as mentioned in the discussion) and my view is that the very strict criteria you have additionally imposed is unnecessary and could result in a biased sample. Similarly, what is the rationale for excluding households that purchased less than 1L of beverages per week per household member? 

7. Flow chart. S1 Appendix. In the last box, I think it would be clearer if you changed "no more than" to "less than or equal to".

8. Income. On page 7, you state the categories for household income. In Table 1 the same categories are labelled "Annual income of the main householder". Is it household income or income of the main panel member? Please check tables and text and make sure they are consistent and correct.

9. Table 1. It would be helpful if you included a footnote to describe the occupational social grade of the main shopper. 

10. Table 1. Under beverages purchased I suggest changing "total" to "all" or "all households" as total implies the associated value is a sum, but this is a median of all households. 

11. Labels of the 7 latent classes. There doesn't appear to be consistency in the way the 7 latent classes are labelled. For example, for Class 2 (juices, low diet and water) the probability of being in the lowest tertile of diet beverage purchasing is 0.65. For class 7 (water) the same probability is 0.64, but this is not labelled "water, low diet and low alcohol". I appreciate the advantages of using simple labels, but then class 2 could just be called "juices" if class 7 is "water". Also, class 6 is labelled "low calories" but for every other class 'low calories' is referred to as 'diet' and the use of 'low' for class 6 is confusing because for all other classes 'low' means less of something, and class 6 is characterised by high purchasing of diet drinks. 

12. Pg 11. The sentence beginning "Conversely, households in class "Alcohol, low SSBs & juices…" - I think it should say 'energy' not 'food'.

13. Table 5. The total energy purchased per household member appears to be quite low, and indicates a substantial (3/4?) of food supply into each household is missing. In the discussion you note that 25-39% of food and beverages out-of-home purchases, but it seems there is still some purchases missing. Do you have any thoughts on this discrepancy?

14. At the end of Table 2 you have absolute energy from purchased beverages. In Table 5 you present percentage of energy from purchased food groups. I wonder if you could present absolute energy and percentage of energy from purchased beverages and food groups. I think the absolute energy purchased is interesting, but also in your discussion and conclusion you state that a greater proportion of purchased energy comes from sweet snacks when compared to SSB but I don't think you have explicitly shown this in the results, although I think that it is true. 

15. 1st paragraph of the discussion. You state that in Classes 1 and 3, purchasing of energy is higher than other classes; however, I don't think they have higher purchasing of energy than the diverse class. 

16. Discussion, pg 12. When you say "healthier baskets" are you referring to shopping baskets/trolleys?

Kathryn Bradbury, Senior Research Fellow, University of Auckland, New Zealand. 

Reviewer #3: Summary

Thank you for the opportunity to review PLOS Medicine D-20-00344R1, "Pattern of beverage purchases amongst British households: A latent class analysis." This manuscript applied latent class analysis (LCA) to identify groups of beverage purchasers using data from about 6,000 households participating in the UK Kantar household purchase panel. Main strengths of the paper include the objectively-collected, longitudinal purchase data and consideration of multiple aspects of purchase behavior (e.g., beverage purchases, nutrient purchases per 1000 kcal, etc.). 

In my read, the paper has two key limitations.

1. The paper would benefit from additional explanation on why LCA is the appropriate method for addressing the research question. 

It was not clear how your approach and results uniquely inform policymaking in the UK or elsewhere. I appreciated the Abstract's statement that we need additional research to understand whom will be most affected by SSB policies. However, it was not clear to me how LCA helps advance this understanding. Why not examine demographic characteristics of high SSB consumers - why does an LCA need to be conducted? If the full pattern of purchases is of interest, why not examine cross-price elasticities across key categories of products? The justification for LCA in relation to policymaking was not clear to me. 

2. The paper needs additional explanation for how these results can uniquely inform policymaking or practice.

The discussion should clearly state how policymakers or practitioners can use these results, if at all, to inform behavior change efforts. The discussion summarized the results well, but did not provide adequate explanation for why the results matter. It would strengthen the paper to provide this explanation, ideally for each main point covered in the discussion.

The main real-world implication offered is that these results reinforce that fiscal policies for obesity prevention should extend to sweet snacks. I don't disagree, but I wasn't sure why an LCA of beverage purchasing is an appropriate approach for understanding whether fiscal policies should extend to sweet snacks. Indeed, a better approach for understanding that question would seem to be the simulation modeling paper cited in reference 39. 

In addition to these important limitations, I have also noted the following suggestions for each section of the paper.

Abstract

3. "…potentially compromising the effectiveness of these policies." I would strike. I very much agree we need to understand who is most affected to understand distributional effects of policies, but it's not clear that lacking this information undermines the policy's effects on behavior. (Won't the policy continue to raise prices and spur reformulation regardless of our understanding of who is affected?). 

4. "…what kinds of fiscal policies might have the most impact on energy intake across this profile." I would strike. As noted above, this paper does not quantitatively examine what kinds of fiscal policies would have the most impact on energy intake. The discussion speculates on this question, but answering this question is not the core of the analysis and thus this should not be stated as an objective. 

5. "households from the classes with higher total spend on SSBs had higher total energy purchases" - typo I think. Should be "total spending"?

6. "Fiscal policies to tackle SSB consumption may have differential effects depending on underlying household beverage purchase behaviours, which may negate or reinforce the policy objective to reduce sugar consumption and obesity." I don't disagree, but I do not think these results can speak to whether fiscal policies have differential effects on households, as no such effects were evaluated or simulated. I would replace this sentence with information on what these results uniquely tell us about policymaking.

Introduction

7. "The two-tiered nature of the levy has encouraged the industry to reduce the sugar content of beverages, resulting in a decrease in the volume of sugars sold in beverages by 30% between 2015 and 2018." I don't disagree that the two-tiered nature of the levy may have encouraged reformulation; however, the construction of this sentence makes it seem like reformulation was the only driver of reduced beverage sugar purchased, with no role for consumer behavior change. Is that accurate? Please revise as needed.

8. "However, understanding whether fiscal policies might be successful at reaching target populations and contribute to reducing SSB consumption and obesity prevalence remains a challenge." I think "contribute" should be "contributing" (same tense as "reaching"). Hard to follow this sentence.

9. I appreciated the brief explanation of LCA as a method to reduce analytic complexity. It would strengthen the paper to explain why LCA is the preferred approach for the earlier-stated research objective of understanding whether fiscal policies might be successful at reaching target populations. For example, why not conduct a simulation model disaggregated by key population group? Or examine cross-price elasticities within key groups? 

10. In the sentence about the benefits of purchase data, you might briefly note that these data do not capture food purchases from restaurants. (I know it's mentioned elsewhere, but this is a key limitation that needs to be stated alongside the key benefits of using purchase data). 

Methods

11. I would strike "representative" in the description of the Kantar data since you were not able to use sample weights.

12. Please include the % of products for which you needed to copy nutritional values across similar products. Likewise the % for which you needed to use the category average. 

13. I would suggest Figure S1 be in the main text to visually depict to readers the important point that the final sample is only ~1/5 of the starting sample.

14. The exclusion of households that don't regularly buy beverages was attributed to "modeling issues." This exclusion reduces the sample size by about 40% from what it would be including those HHs, so would be nice to have more understanding of why these HHs couldn't be included. 

15. I was curious about the exclusion of households with +/-3 SDs of beverage purchases. Why was this exclusion needed if you were using tertiles of purchases? Additionally, it was somewhat surprised that this exclusion resulted in a loss of about 20% of the sample. It would seem that much fewer than 20% of the sample would have values this extreme. But perhaps this restriction was applied to each beverage group, resulting in loss of different households for each beverage group? Some clarification would help. Additionally, given that this exclusion results in loss of quite a few households, additional justification should be provided as to why this exclusion is important and does not bias the results. For example, it would seem that excluding households at the low end (-3 SDs) on purchases would result in loss of key groups of HHs who are truly "low" purchasers of a given category. Also, I'm guessing that purchases are nearly always right skewed, so those at -3 SDs probably have 0 purchases of the products, which is not an implausible or unrealistic value for many households to have. (My household never buys SSBs, for example). Why does it make sense to exclude these HHs?

16. In the measures section, would help to very briefly define SSBs. Same for fruit juices and milk-based juices. What are milk-based juices? May also help to clarify whether these would be subject to the UK SDIL. 

17. Relatedly, I was curious that SSBs were not separated into higher and lower-sugar beverages that would be taxed differently. Can you explain the rationale for this decision? 

18. Again, would help to explain what the "modeling issues" were that required examining beverage volume by tertile. Why not quartile? Quintile? 

19. "Normal" BMI cutoffs - suggest instead referring to WHO. "Normal" cutoffs in Asia are not the same as in UK. 

Results

20. "In addition to being classified in the Diverse class, households purchasing high volumes of SSBs were split…" This was confusing - made it seem like those in the Diverse class would further subdivided, when in fact I think you meant to refer to two other classes. The classes were not (based on my understanding) nested within one another, correct?

21. Throughout: Suggest using "person first" language. For example, several sentences refer to "obese respondents." I believe the recommended language is "Respondents with obesity" (as used elsewhere in the results). 

22. How was the decision made to present the RRRs with the diverse class as the referent? I found it hard to know what to make of these results. Does the diverse class have some inherent meaning with regard to policymaking?

Discussion

(See above notes about highlighting what this paper uniquely adds to the literature).

23. The paragraph beginning "Our result suggests that fiscal policies to tackle SSB consumption may have differential effects…" I don't disagree with the sentiment here, but I wasn't sure how these results speak to that conclusion. The paper did not estimate or simulation how purchases would change among HHs in each beverage purchasing class, but rather describes current purchases. I would suggest revising this paragraph.

24. "…proportional greater impact on households at highest risk of obesity." Again, not sure that this paper can speak to which households would be most impacted given behavior change was not modeled. Additionally, households are not at risk of obesity, individuals are - suggest revising. 

25. Limitations paragraph, "Patterns of beverage purchasing might different from those made to take-home." Couldn't quite follow this sentence. 

Misc/Throughout:

26. Given that this paper used purchase data, not dietary intake data, I would suggest avoiding referring to "diet" or "consumers" throughout. Instead, "purchases" and "purchasers." For example, "Households with less healthy diets overall" is inaccurate; diets were not assessed. 

27. Tables did not fit onto page; formatting should be revised. 

28. Could the LCA results feasibly be shown as a figure? Might help make the results more digestible to readers. 

29. Figure S1 (flowchart) - Please add boxes for number excluded at each stage, not just number remaining. (Reader should not be asked to do the math on these; exclusion flowcharts should show both number retained and number excluded for each step).

[LINK]

---

## [Decision Letter · Decision Letter 2]

17 Jun 2020

Dear Dr. Berger,

Thank you very much for re-submitting your manuscript "Patterns of beverage purchases amongst British households:a latent class analysis" (PMEDICINE-D-20-00344R2) for review by PLOS Medicine.

I have discussed the paper with my colleagues and the academic editor and it was also seen again by reviewers. I am pleased to say that provided the remaining editorial and production issues are dealt with we are planning to accept the paper for publication in the journal.

[LINK]

We look forward to receiving the revised manuscript by Jun 22 2020 11:59PM. 

Sincerely,

Adya Misra, PhD

Senior Editor 

PLOS Medicine

plosmedicine.org

Requests from Editors:

Abstract

Could you perhaps mention briefly if the household data can be considered representative of the British population?

Is “low occupational social grade” commonly used? I would perhaps suggest socioeconomic status here in the abstract and also on page 27

On page 18 and throughout- please provide exact p-values, unless p<0.001. Please provide these throughout the main text and tables, as needed

Please provide 95% confidence intervals throughout 

Please remove financial disclosure and COI from page 2 as imported from EM 

Comments from Reviewers:

Reviewer #2: I have read through the authors' response and the revised paper. I would like to thank the authors for their engagement with my (and the other reviewers') comments. The authors have provided comprehensive responses, and have substantially altered the manuscript. I now think that the rationale is stronger, and what their study shows and what it cannot show is also clearer. Loosening the inclusion criteria to include more households is also a positive step. Overall I think the paper is much improved and I have no further comments. I would also like to thank the authors and journal for the opportunity to review their paper - the author responses to my comments have made me think and understand more deeply how these commercial datasets can be used, as well as their limitations.

[LINK]

---

## [Editor Report · Decision Letter 3]

3 Aug 2020

Dear Mr. Berger, 

On behalf of my colleagues and the academic editor, Dr. Sanjay Basu, I am delighted to inform you that your manuscript entitled "Patterns of beverage purchases amongst British households:a latent class analysis" (PMEDICINE-D-20-00344R3) has been accepted for publication in PLOS Medicine. 

PRODUCTION PROCESS

PRESS

PROFILE INFORMATION

Thank you again for submitting the manuscript to PLOS Medicine. We look forward to publishing it. 

Best wishes, 

Adya Misra, PhD

Senior Editor 

PLOS Medicine

plosmedicine.org